# Usage of Object Matching Algorithms Combined with Mixed Reality for Enhanced Decision Making in Orbital Reconstruction—A Technical Note

**DOI:** 10.3390/jpm13060922

**Published:** 2023-05-31

**Authors:** Max Wilkat, Nadia Karnatz, Felix Schrader, Lara Schorn, Julian Lommen, Aida Parviz, Henriette Louise Möllmann, Majeed Rana

**Affiliations:** Department of Oro-Maxillofacial and Facial Plastic Surgery, University Hospital Düsseldorf, Moorenstr. 5, 40225 Düsseldorf, Germany

**Keywords:** decision making, mixed reality, surface matching, volume matching, patient-specific implants, patient education

## Abstract

This technical note describes the usage of object matching to virtually compare different modes of reconstruction in orbital trauma and display the results to the surgeon and the patient pre-operatively via mixed reality devices for enhanced surgical decision making and immersive patient education. A case of an orbital floor fracture is presented for which surface and volume matching were implemented to compare orbital reconstruction utilizing pre-fabricated titanium meshes versus patient-specific implants. The results could be visualized by mixed reality devices to further enhance surgical decision-making. The data sets were demonstrated to the patient in mixed reality for immersive patient education and enhanced shared decision making. The advantages of the new technologies are discussed in view of the new possibilities of improved patient education and informed consent processes, as well as new ways of teaching medical trainees.

## 1. Introduction

Orbital floor fractures are common consequences of facial trauma [1,2]. The complex three-dimensional geometry of the orbit and the limited surgical accessibility still represent a challenge for treatment. It is known that precise reconstruction of the orbit true to the anatomical conditions is of foremost importance to re-establish function and aesthetics [3,4,5]. While autogenous modes of reconstruction show no predictable result, the use of pre-fabricated and intra-operatively modified titanium mesh implants has been established as a standard procedure for orbital reconstruction. The use of computer-assisted design and manufacturing of individual patient-specific implants (PSIs) was a revolutionary breakthrough in craniomaxillofacial surgery and orbital repair. By mirroring the healthy, unaffected site to the orbital defect preoperatively on the basis of the DICOM data set with appropriate software, a PSI can be produced using the voxel-based data set and the standard CAD/CAM procedure [6,7,8]. The use of PSIs allows precise reconstruction, minimizes the risk of postoperative enophthalmos and diplopia, and facilitates a shorter surgery duration [9,10]. However, this mode of reconstruction comes with disadvantages, such as delayed repair due to the duration of the individualized manufacturing process and higher costs compared to pre-fabricated titanium meshes. That is why the need for a tool arises, which allows one to differentiate between those cases that would profit from a PSI and those for which a PSI would probably pose an over-treatment with an unnecessary delay in surgery. Here, we present the use of new software algorithms allowing us to virtually measure the ideal positioning of pre-fabricated titanium meshes and compare the achievable reconstruction results of different modes of reconstruction. The data can be visualized by mixed reality (MR) devices to further enhance surgical decision making.

However, communicating this complex surgical procedure to patients is the next obstacle to overcome. If the patient has sufficient knowledge about the procedure and the risks involved, informed consent can be obtained [11]. A 3D visualization of medical data like CT scans as well as the surgical procedure with MR technology can be a tool to enhance patient information. For example, advanced photorealistic three-dimensional renderings based on tomographic data can support patients in their decision making through the spatial visualization of raw data that would otherwise be difficult to understand [12]. Furthermore, MR technology can be particularly helpful as a visual interface between sectional imaging and the explanation of surgical planning in terms of understanding. MR technology offers a multimodal interactive user interface to enhance communication between doctor and patient [13]. The use of this new technology might enhance and support better patient education, leading to greater transparency and adherence to the treatment.

## 2. Methods and Results

### 2.1. Case Demonstration

A patient with an orbital floor fracture due to a fall was in need of primary orbital reconstruction. He suffered from binocular double-vision, hypesthesia of the right infraorbital nerve, and a hypoglobe with slight enophthalmos on the affected right side (see Figure 1). The CT data showed a defect size of 17.9 mm in the coronal plane and 23.4 mm in the sagittal plane. Due to the defect size, the mode of reconstruction was chosen as an alloplastic reconstruction with a titanium mesh. To show the different fitting qualities, a pre-fabricated titanium mesh was compared to a PSI.

### 2.2. Planning Procedure

Virtual surgical planning involved loading the CT data set in the axial plane and bone window with a slice thickness of 0.75 mm into the software Brainlab Elements Contouring 4.5’ (Brainlab, Munich, Germany). After virtual mirroring of the healthy, unaffected site to the fractured site, the defect area was outlined. A standard pre-fabricated titanium mesh of the KLS Martin osteosynthesis system for orbital reconstruction was loaded into the data set from the software library. The titanium mesh was put into place by the option ‘object matching.’ This algorithm allows the use of two different methods: surface and volume matching. For surface matching, a point cloud is matched onto a given surface (for volume matching into another point cloud) by rotating and translating the point cloud to minimize the root mean square (rms) distance of the points to the surface (or volume, respectively). Surface matching is usually recommended for repositioning bones, and volume matching is usually recommended for replacing bones (e.g., after tumor resection). From a surgical point of view, orbital reconstruction allows repositioning of bone only in smaller defects that show small, displaced fragments still pedicled to the periosteum, while bigger defects must be reconstructed by replacing the lost bone fragments. That is why, in our experience, volume matching delivered better results as it aimed for a matching into the volume of the mirrored floor, thereby re-contouring the fractured orbit (see Figure 2). Surface matching, on the other hand, places the implant onto the mirrored floor, thereby over-contouring the orbital outline.

The reconstruction result of the virtually positioned pre-fabricated titanium mesh was compared to the possible reconstruction result of a PSI. Therefore, the PSI design workflow was followed as described before [14,15]. The DICOM data set was sent to KLS Martin GmbH (Tuttlingen, Germany) via their online platform, IPS Gate. Designing steps were reviewed via online chat, including the definition of the outline of the PSI following the contour of the virtually reconstructed orbit and adding design features like drainage holes and navigational groves. The STL file of the final geometry of the PSI was provided by KLS Martin. The STL file was imported into the Brainlab data set, allowing direct comparison (see Figure 2).

### 2.3. Decision Making and Patient Education

For visualization, we used the Magic Leap Device, driven by the Brainlab Elements software package. This mixed reality device can project 3D objects via semi-transparent glasses into the field of view of the carrier and allow interaction via a motion-sensing pointer device. The projected object can be freely inspected by the viewer by either moving it in three-dimensional space with the help of the pointer device or by freely moving around the projection via locomotion. By stepping through the object, a cut-through plane is automatically generated, which allows an intersection view in different planes depending on the viewing angle of the object. Thereby, an immersive inspection of the planning data displaying the different modes of reconstruction of the patient’s individual anatomy and fracture type by the patient himself was possible. The demonstration of the data sets revealed a mismatch of the pre-fabricated titanium mesh with a distance of 1.5 mm to the mirrored floor and a gap of 1.5 mm to the posterior ledge. Compared to the PSI, which adapts perfectly to the reconstructed anatomical shape as it was designed that way, the usage of the pre-fabricated titanium mesh would pose the need for intra-operative adaptation to the patient’s anatomy, as further pointed out by the virtual comparison (see Figure 3).

The whole data set was also presented to the patient via the MR device. The demonstration of the fracture pattern, defect size, and different modes of reconstruction greatly enhanced the patient’s education. As it was decided to use a PSI, arguments for this surgical decision could be visualized for the patient, enhancing their understanding and leading to greater compliance. The advantages of an easier, more achievable, and more predictable reconstruction result as demonstrated via MR could be countered by the disadvantages of delayed surgery and higher costs, which have been made fully transparent to the patient.

### 2.4. Surgical Procedure and Clinical Outcome

After decision making and choosing the PSI as mode of reconstruction, the finished geometry was manufactured into a PSI out of 0.4 mm thick titanium alloy (Ti6Al4V Grade IV) via selective laser melting.

The surgical procedure involved preparation of the fractured orbital floor after a transconjunctival approach (see Figure 4). The PSI could be inserted and fixed to the orbital rim with one 4 mm long osteosynthesis screw. The correct implant position was checked intra-operatively using trajectory-guided real-time navigation. A post-operative CBCT scan confirmed correct positioning (see Figure 5). Postoperatively, the patient showed no enophthalmos, double vision, or impairment of orbital mobility. Implant placement proved to be accurate and fast.

## 3. Discussion

### 3.1. Surgical Decision Making

Modes of orbital reconstruction have been vastly investigated, and many studies have shown the benefits of the use of PSIs, resulting in accurate anatomical reconstructions with high predictability, shorter surgery duration, and overall high-quality outcomes. However, the use of PSIs poses disadvantages such as a delay in surgery due to the prolonged manufacturing process and higher production costs. That is why their usage is not standard procedure in many countries yet. Regardless, there are cases that clearly profit from this procedure, and the challenge remains to identify these cases.

Here, we present new software algorithms allowing the match of a pre-fabricated titanium mesh into the mirrored and reconstructed orbit to evaluate the primary fitting and estimate the extent of intra-operative adjustments as a tool for pre-operative surgical decision making.

The decision to surgically treat an isolated orbital fracture is mainly based on clinical findings such as binocular diplopia, aesthetically disturbing globe malpositioning such as enophthalmos, or hypesthesia of the infraorbital nerve. There are many studies investigating the different clinical outcomes of conservative treatment versus surgical treatment. While some studies found that certain clinical findings have the possibility of spontaneously resolving and therefore advocate conservative treatment, others found that spontaneous symptom resolution depends on the defect size and area, thereby defining defect-driven indications for surgical treatment [16,17,18,19]. Once the indication for surgery has been set, the question arises as to which kind of reconstruction should be used. There is a wide variety of possibilities for orbital reconstruction: calvarian bone grafts or other autogenous tissue grafts and different alloplastic materials—forming stable versus flexible, re-absorbables versus permanent materials, pre-fabricated versus patient-specific—involving materials such as PDS, PEEK, or titanium implants [20,21,22]. There are many studies investigating the different modes of reconstruction, pointing out various advantages and disadvantages [4,15,23]. However, a clear indication for a certain material has not been defined. While recent opinions support the idea that the bigger the defect, the more stable and permanent the reconstruction material should be, the definition of implant geometry has also been emphasized. However, there is no clear indication at which point PSIs should be utilized, and eventually the decision is made by the treating surgeon. Therefore, we present a tool to help in surgical decision making [11,12,13].

### 3.2. Patient Education and Shared Decision Making

Once the surgical decision has been made, it must be communicated to the patient, which can pose a challenge of its own at times. The discrepancy regarding a patient’s and a physician’s understanding of appropriate communication of information has been examined in various studies. This discrepancy in understanding involves technical and/or medical terms and a lack of knowledge concerning human anatomy and surgical procedures [24,25,26]. However, giving consent to a surgical procedure does not necessarily mean that the patient has understood their diagnosis, planned therapeutical measures, or the ability to assess potential risks [27]. Finding a common communicative level in the physician-patient relationship is intricate. The visualization of information enables a new innovation in patient–physician communication [28]. MR-supported demonstrations of surgical interventions represent an alternative and advanced technique of patient education as opposed to conventional methods. It offers insights into three-dimensional anatomical structures and increases a patient’s understanding despite a lack of prior knowledge. The main advantages of the new type of technology are the immersive visualization of the patient’s own pathology and anatomical features, as well as a language barrier-free and interactive form of communication [29]. MR technology enables the creation of immersive 3D models of the patient’s facial anatomy, including the fractured orbital floor. This allows healthcare professionals to visually explain the injury, the surgical procedure, and the expected outcomes. Patients can gain a better understanding of their condition, reducing anxiety and improving satisfaction [30].

By overlaying the patient’s preoperative imaging, such as CT scans, onto the MR environment, surgeons can demonstrate the planned surgical approach and simulate the reconstruction process. Patients can actively participate in the surgical planning process by visualizing potential outcomes and discussing alternative options. This immersive experience aids in managing expectations, addressing concerns, and facilitating the informed consent process [31]. This fosters shared decision-making and patient empowerment [32]. MR technology also enables personalized risk assessment by simulating potential surgical complications specific to the patient’s case. Patients can visualize the likelihood and consequences of complications, aiding in their decision-making process [33]. Interactive modules and simulations allow patients to explore and comprehend complex medical information at their own pace. This enhances patient education, facilitates discussions, and improves compliance. MR technology can even facilitate remote consultations between patients and healthcare professionals, bridging the gap of physical distance, which might likely apply in secondary cases where patients are in worldwide search of a specialized expert in their medical field. Patients can have virtual consultations, view 3D models, and discuss surgical plans with the surgeon, even when they are not physically present. This improves accessibility, reduces travel burdens, and promotes patient-centered care [30].

MR technology holds great promise for improving patient communication and enhancing the informed consent process for patients requiring orbital floor fracture surgery. Through immersive visualization, interactive planning, and realistic simulations, MR empowers patients and facilitates shared decision-making. This overall advancement in knowledge and understanding of individual diseases results in higher patient satisfaction, thus resulting in better patient compliance [34,35]. As different studies have already shown that the additional use of visualization can benefit patients regarding preoperative anxiety levels and general knowledge about the disease, we share the opinion that MR techniques lead to higher standards of patient education as well as patient consent [36,37].

Challenges in implementing MR technology for orbital floor fracture surgery include technical limitations, cost-effectiveness, data privacy, and the need for standardization. Future research should focus on validating the efficacy of MR in improving patient outcomes, optimizing user interfaces, and integrating MR with other healthcare systems. With all the mentioned expensive technologies at hand, prospective studies must elucidate their clinical benefits to prevent over-treatment and ensure an effective price-performance ratio to help maintain a viable healthcare system.

### 3.3. Medical Education

The surgical repair of orbital floor fractures requires precise anatomical knowledge, technical skills, and an understanding of surgical approaches. Traditional teaching methods may have limitations in providing comprehensive training experiences. The extensive data sets gathered during the virtual surgical planning of actual patient cases and their immersive visualization via MR technology offer an innovative educational approach to enhance the learning process for medical students and specialty trainees.

MR enables medical students to visualize the complex anatomy of the orbital floor in a three-dimensional environment. They can explore detailed anatomical structures, enhancing their understanding of the fracture and the surgical repair process. This immersive visualization promotes spatial awareness and improves anatomical knowledge retention [38]. MR technology also allows young residents during their specialty training to engage in interactive virtual simulations of surgical procedures for orbital floor fracture repair. They can manipulate virtual instruments, practice surgical techniques, and experience realistic surgical scenarios. These simulations provide a safe and controlled environment for skill development and decision making, fostering confidence and competence [39].

More advanced MR systems can even incorporate haptic feedback, providing medical trainees with tactile sensations during virtual surgical procedures. This feature allows them to experience the resistance, texture, and force feedback associated with surgical maneuvers. The inclusion of haptic feedback enhances the realism of the training experience and improves psychomotor skill development [40]. Additionally, step-by-step guidance and instructional overlays during surgical simulations can be provided. Medical trainees can follow visual cues, annotations, and audio instructions, enhancing their understanding of the surgical steps involved in orbital floor fracture repair. This real-time guidance facilitates skill acquisition and reduces the learning curve [41].

MR facilitates collaborative learning experiences by enabling medical students and specialty trainees to interact with virtual patients, surgical mentors, and peers in a shared virtual environment. They can engage in team-based discussions, receive real-time feedback, and participate in collaborative surgical planning. This promotes interdisciplinary collaboration, communication skills, and a sense of community among medical learners [41].

Challenges in integrating MR into surgical education include the cost of hardware and software, technological limitations, validation of training outcomes, and the need for faculty training. Future research should focus on assessing the effectiveness and transferability of MR-based training methods, optimizing user interfaces, and developing standardized curricula.

MR technology presents a novel and promising approach to surgical education for orbital floor fracture repair. By providing immersive anatomical visualization, interactive virtual simulations, and practical skill acquisition, MR can enhance the learning experience for medical trainees. Overcoming challenges and continued research efforts will further optimize MR-based surgical education, ultimately benefiting the future generation of surgeons.

## 4. Conclusions

The demonstrated matching algorithms further enhance virtual surgical planning, which can especially help with surgical decision making as there are no clear guidelines for the optimal mode of orbital reconstruction. Using the MR platform as a common form of communication, treating physicians are enabled to not only evaluate their own treatment plans in an immersive environment but also communicate more intensely with their patients, and, thus, represents an important tool in the shared decision-making process. Moreover, medical education could be greatly enhanced by the use of these new tools in view of the difficulties in surgical decision making.

## Figures and Tables

**Figure 1 jpm-13-00922-f001:**
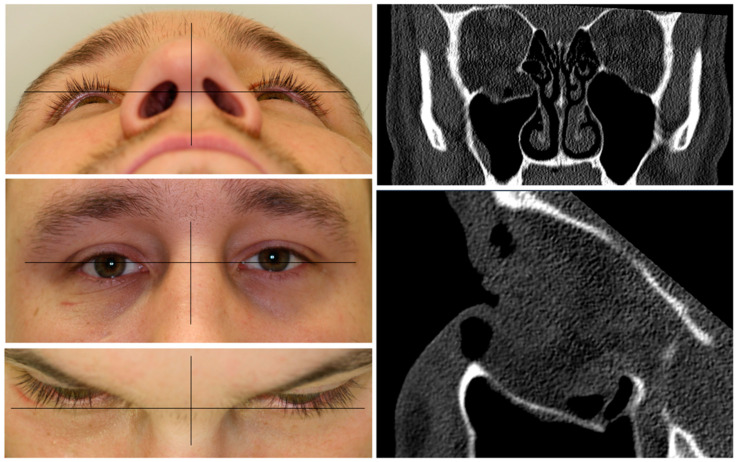
Clinical photographs of the patient showing an enophthalmos in the perspective from below and above, with a hypoglobe present on the right affected side as visible in the frontal view. The coronal and sagittal planes of the pre-operative CT scan displaying the traumatic defect of the orbital floor on the right side.

**Figure 2 jpm-13-00922-f002:**
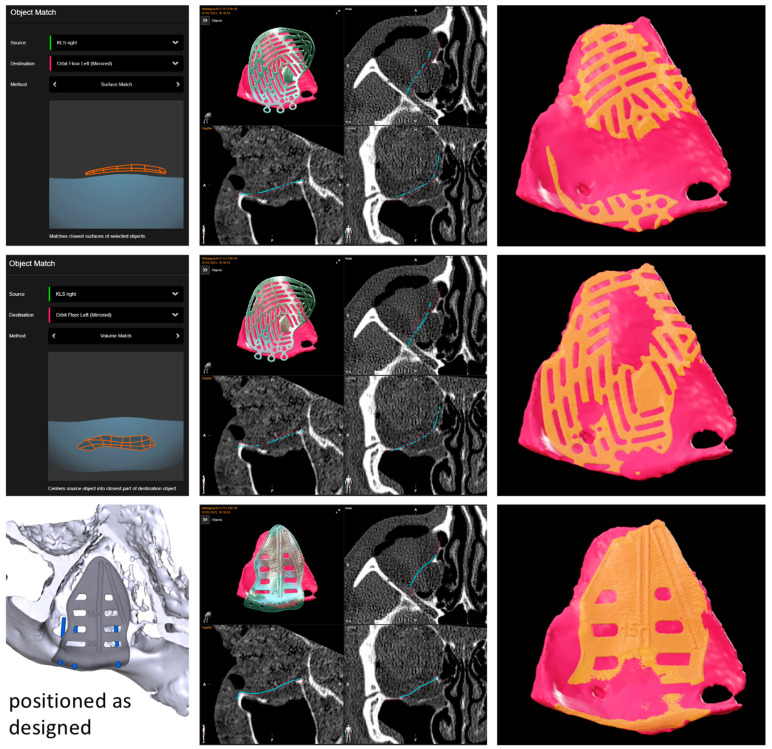
Results of the object matching: the upper and middle rows showing the pre-fabricated titanium mesh put in place by surface (upper row) and volume (middle row) matching, while the lower row displays the geometry of the PSI design in its planned position. The right column displays the inter-sectioning volume (orange) of each reconstruction result with the mirrored orbital floor (pink). While the inter-sectioning volumes for the pre-fabricated titanium mesh were 0.07 cm^3^ for surface matching and 0.12 cm^3^ for volume matching, the inter-sectioning volume of the PSI was the biggest, with 0.14 cm^3^ offering the reconstruction result most true to the mirrored anatomy. Moreover, the areas of the supporting bone surfaces on which the titanium meshes rested were significantly smaller. This also applied to the distances between the supporting bone surfaces and the different titanium meshes, which showed no gap at all for the patient-specific implant.

**Figure 3 jpm-13-00922-f003:**
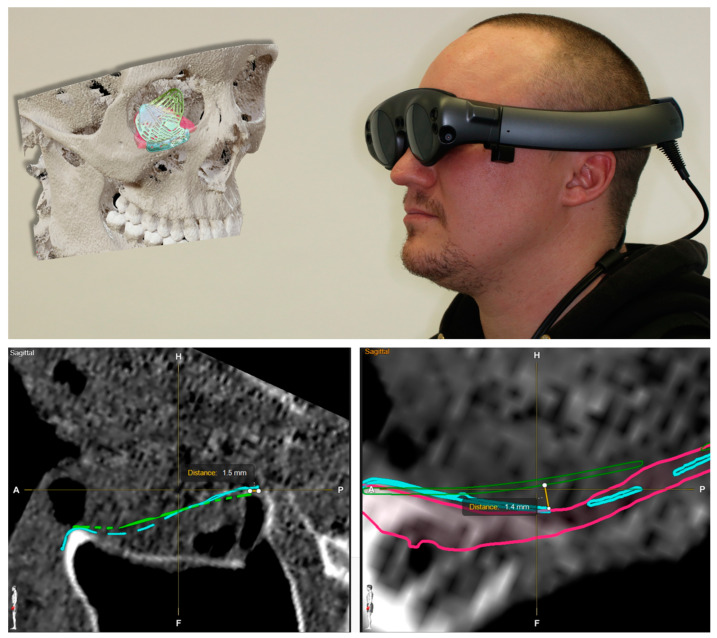
Demonstration of the data set to the patient via mixed reality device Magic Leap. A mismatch with distances of the pre-fabricated titanium mesh to the orbital floor of 1.4 mm and a gap of 1.5 mm to the posterior ledge reveal the need for intra-operative adaptation for a perfect fit compared to the PSI.

**Figure 4 jpm-13-00922-f004:**
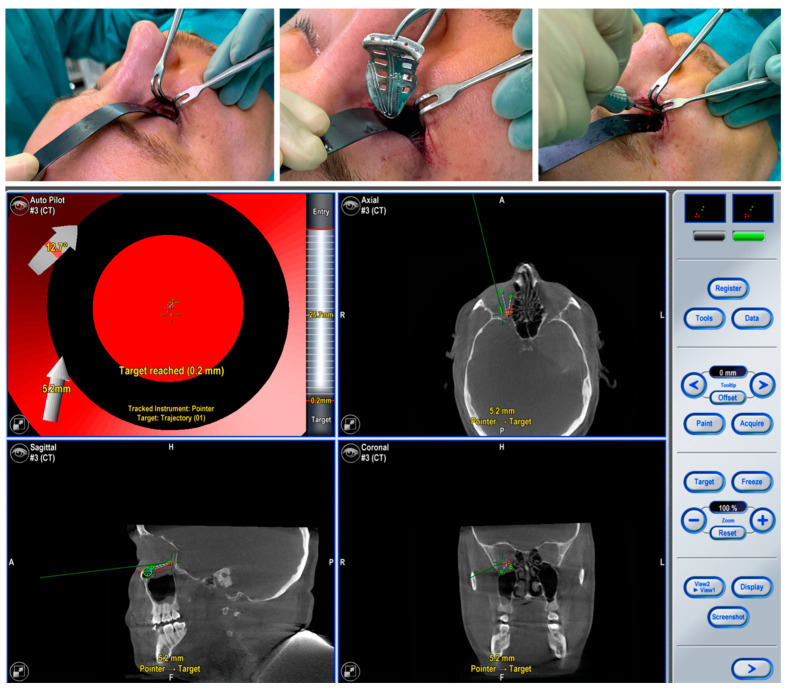
The intraoperative photographs show the preparation of the left orbit via a transconjunctival incision and the insertion of the patient-specific implant, which is fixed with one 4 mm osteosynthesis screw at the infraorbital rim. Intra-operative real-time navigation using a navigation probe to follow the grooves on the implant confirms the correct implant positioning by reaching the target of the set trajectories.

**Figure 5 jpm-13-00922-f005:**
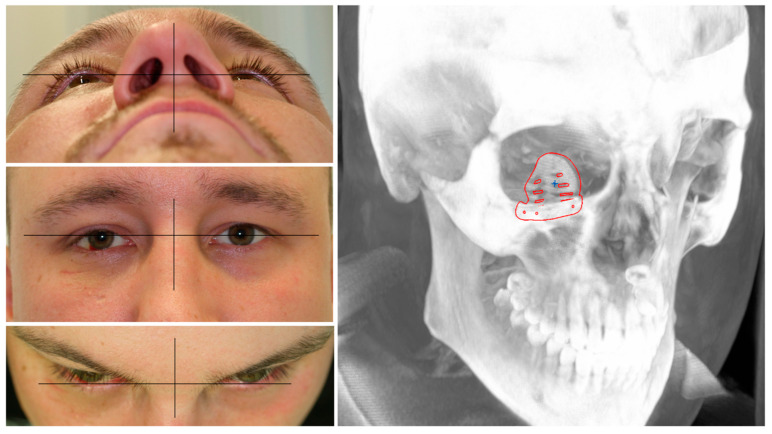
Post-operative clinical photographs show an improvement of the enophthalmos and hypoglobe. Merging the pre-operative planning CT data with the post-operative CBCT scan confirms correct positioning as the virtually planned patient-specific implant co-localizes with the inserted implant.

## Data Availability

The datasets used and/or analyzed during the study are available from the corresponding author on reasonable request.

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
