# Peer review of "Usage of Object Matching Algorithms Combined with Mixed Reality for Enhanced Decision Making in Orbital Reconstruction—A Technical Note"

_jpm, 2023, doi:10.3390/jpm13060922_

Round 1

Reviewer 1 Report

I have read with interest the paper "Usage of object matching algorithms combined with mixed reality for enhanced decision making in orbital reconstruction – a technical note".
It is a technical note on the use of mixed reality for decision making and patient communication  in orbital fracture reconstruction. 
The technical approach is very 
sophisticated and the paper is very clear to the reader, tough concise.
Considering the extended use of expensive technologies, I suggest that a consideration is given in the Discussion to the need of prospective studies on the clinical benefit of the different technologies proposed (i.e. PSI vs prefabricated meshes). 

Reviewer 2 Report

This paper has great clinical significant. 

Minor 

1) Please change " PSIs (Page1 Line42)" to "Patient specific implants (PSIs)"

2) In discussion, please add not only the education for patients but also the content of education for students.

Reviewer 3 Report

The article is written accurately and has innovation.
